# Thalamic input to the lateral amygdala determines the temporal window of fear-memory association
Natália Madeira[1,2], Inês Campelo[1] & Rosalina Fonseca [1,2] ✉

Memory consolidation is highly influenced by ongoing experiences. Here, we explore the temporal rules that determine whether events are cooperatively associated or competitively separated. We show that neutral events are associated with fearful events if they occur within less than 30 min. In some individuals, memory association can lead to a competitive suppression of the fearful response by the neutral event. Activation of the thalamic MGm inputs to the lateral amygdala, results in an increase in memory association, whereas manipulation of the cortical inputs have no effect. Introducing a third event leads to competition depending on the temporal relationship between the initial association and the competitive event. Our results show a critical temporal rule of memory association, modulated by thalamic activity that shapes fear memory consolidation.

The ability to acquire and retrieve memories is fundamental to adequate behavior to environmental challenges[1]. However, memories are not immutable traces but rather constantly changing representations[2]. Indeed, even if each learning event can be seen as a "new" memory formation process, considerable evidence indicates that new memories are formed in an interleaved fashion, upon a large network of pre-existing knowledge[3,4]. This implies that events occurring at different time points can be associated or linked and remembered as a single memory trace[5,6]. This cooperative effect of different events in memory acquisition is also observed by exposing animals to a novel environment, which acts as a positive reinforcer of memory[7]. These results show that distinct events, including an unexpected novel event, can be associated in time, promoting cooperative memory consolidation. However, as important as associating events in time, the ability to distinguish events is highly relevant. Even more if one considers aversive events in which the inability to distinguish between aversive versus neutral events leads to a generalized anxious response that is maladaptive[8]. Competitive interactions between events would allow selective storage of some events over others, presumably due to their valence[6,9]. The proposed mechanism for memory cooperation and competition involves interactions between overlapping neurons participating in the engram of the interacting events[10]. Given that individuals are exposed to different events constantly, it remains unanswered how time influences memory cooperation and competition and the rules that determine whether different events cooperate or compete.

To address these questions, we developed a modified fear discrimination learning paradigm, in which animals are exposed to two events, a neutral event and a fearful event, separated by defined time intervals. In

auditory fear-conditioning, subjects learn to associate a neutral stimulus (conditioned stimulus - CS) with a coincident aversive stimulus (unconditioned stimulus - US[11,12]). Once conditioned, the neutral stimulus will evoke a conditioned response (CR), resulting in immobilization or freezing in rodents, providing a behavioral measure of the learned association. Conversely, in auditory discriminative learning, the animal learns to distinguish between two neutral stimuli, a CS+ paired with the US and a CS- unpaired. As the animal learns this task, responses of amygdala pyramidal neurons increase upon CS+ presentation and decrease upon CS- presentation[13]. The leading cellular model underlying auditory fear conditioning is a form of Hebbian long-term potentiation (LTP), induced by the association between the auditory thalamic and cortex projections (CS) and the nociceptive input (US)[14]. The auditory information reaches the amygdala via direct projections from the medial division of the medial geniculate nucleus (MGm- thalamic input) and an indirect projection, through the projection of the ventral division of the medial geniculate nucleus (MGv – cortical input) to the auditory cortex which in turn, projects to the LA[15,16]. Previous studies have shown that thalamic and cortical input projections contribute differently to the acquisition of auditory discriminative memories. Cortical projections have been associated with the processing of more complex CSs, whereas the direct thalamic projections have been considered to be a fast but less accurate information pathway[17]. This is supported by the observation that neurons in the MGv and primary auditory cortex are sharply tuned and tonotopically organized, whereas neurons in the MGm show non-tuned auditory responses and are multisensory, responding to other sensory modalities such as visual cues. While the activation of either the cortical or thalamic inputs is sufficient for fear-conditioning learning, in

[1]Learning and Plasticity, i3s – Instituto de Investigação e Inovação em Saúde, Universidade do Porto, Porto, Portugal. [2]These authors contributed equally: Natália Madeira, Rosalina Fonseca. ✉e-mail: rfonseca@i3s.up.pt

auditory discriminative learning, co-activation of both inputs seems to be necessary for discrimination[18].

As for memory, synaptic inputs can also interact cooperatively and competitively[19–21]. Synaptic cooperation is observed when the induction of a long-lasting form of plasticity in one set of synapses, stabilizes a transient form of plasticity in a second independent set of synapses. In synaptic cooperation, plasticity maintenance is achieved by an interaction between input-specific 'synaptic tags' and the capture of plasticity-related proteins (PRPs) synthesized in the soma or local dendritic domains[19]. If PRPs are limited, for example, by blocking protein synthesis, or by increasing the pool of activated synapses, then synaptic competition is observed[20,21]. Synaptic cooperation and competition are two forms of heterosynaptic plasticity based on the interaction between PRPs and activity-dependent synaptic tags. Since maintenance of plasticity depends on the interaction between these two interactive but independent cellular processes, synaptic cooperation and competition allow synapses to integrate cell-wide activity within large time windows[19,22,23]. Taking advantage of the detailed knowledge of the circuits underlying fear memory acquisition, we have built a detailed model of the temporal rules underlying synaptic cooperation and competition between the cortical and thalamic inputs to the lateral amygdala. We found that cooperative maintenance of long-term potentiation in cortical synapses can occur if thalamic synapses are stimulated within a temporal window of 30 min[24,25]. The reverse interaction, in which the thalamic synapses are reinforced by cortical stimulation, is restricted to a much shorter time window of 7.5 min due to the activation of cannabinoid receptor type 1 (CB1R). If CB1Rs are inhibited, thalamic synapses can extend the time interval of cooperation to 30 min, similar to the time interval observed for cortical synapses. Thus, thalamic-cortical cooperation reinforces both inputs in an associative but temporally asymmetrical manner. This suggests that, once plasticity is induced, ongoing activity and activation of CB1R restrict the time window for thalamic-cortical cooperative plasticity. Thalamic and cortical synapses also engage in synaptic competition[26]. By introducing a third stimulating electrode placed on internal capsule fibers to activate a second thalamic input, we could create a competitive load by increasing the number of activated synapses that capture PRPs and as for cooperation, time is crucial to induce competition. If the second thalamic weak stimulation was delayed by 30 min, no competition was observed, suggesting that synapses are susceptible to perturbation by competition for a restricted time window. Inhibition of CB1 receptors promotes competition, possibly by reducing presynaptic glutamate release and synaptic activation[26]. Building on this model of synaptic cooperation and competition between thalamic and cortical inputs to the lateral amygdala, we tested the temporal rules of memory cooperation and competition as well as the contribution of the thalamic and cortical inputs to discriminative fear memory acquisition.

## Results

### Memory cooperation has a restricted time rule
We have found that cortical and thalamic synapses projecting to the lateral amygdala only cooperate if synaptic plasticity is induced within 30 min[25,26]. According to this, we predicted that events occurring within this 30 min time window will be associated, resulting in a form of memory cooperation, whereas longer time windows will allow event separation. To test this hypothesis, we expose animals to two different events separated by three different time points, 7.5, 30 min, and 1 h. Event one consisted of one particular context (1) where an auditory stimulus (CS-, 20 s, 10KHz pips) is presented, whereas in event two, occurring in a different context (2), a second auditory stimulus (CS +, 20 s, 2 KHz tone) is presented co-terminating with a foot shock (Fig. 1). As we expected, we found that for the shortest time interval used, 7.5 min, the events are associated and animals respond with a freezing response to both stimuli, CS- and CS +, in the test session (Fig. 1A). Also, for the longest interval, as we hypothesized, animals can separate the two events and freeze only to the CS+ stimuli (Fig. 1C). The cooperative effect seen in the 7.5 min interval is not due to an overall

generalization of fear, given that, if CS- is not present in the first event, no response is seen in the test trial (Supplementary Fig. 1A). Importantly, there is also no association between the CS- and the US itself. If the CS+ is omitted during the US event, no fear response is observed to the CS- or the CS+ in the test trial, showing that responding to the CS- requires an association with the second auditory cue (Supplementary Fig. 1B). This association does not depend on the particular frequencies used, given that if the two auditory cues are switched, animals still form an associative fear memory and respond with freezing to both aversive and neutral cues (Supplementary Fig. 1C).

For the 30-min interval, the average freezing response is similar to the one observed in the 7.5-min with significantly higher freezing levels for both CS- and CS+ compared to baseline responses (Fig. 1B). This is also evident in the analysis of the discriminative index shown in Fig. 1D. For the time interval of 1 h, the discriminative index is significantly higher than for the other time intervals tested, showing that in this case, neutral and aversive events are distinguished. Interestingly, when analyzing individual responses, we found that for the intermediate time interval (30 min), animals were distributed in two distinct groups. One group of animals freezes to both stimuli, CS- and CS +, showing association, whereas a second group responds significantly less to both cues. This is better appreciated by plotting the freezing levels to CS+ and CS- of individual subjects, as shown in Fig. 1E. For the short (7.5 min) and long interval (60 min) there is no correlation between CS+ and CS- freezing levels, with values distributed around higher CS- values in the case of the 7.5 min (red line) and low CS- values for the 1-hour interval (green line). In the case of the 30 min, we found a significant correlation (blue line) between CS+ and CS-, representing the two groups of responses, higher freezers, and lower freezers to both cues. This observation suggests that animals in the 30-min interval associate the two auditory cues but respond either as aversive or as neutral to both cues. Interestingly, this association is already present if tested shortly after the training session (3 h; Supplementary Fig. 2A) and it is not transient, being observed even if animals are tested 72 h after training (Supplementary Fig. 2C).

In our synaptic analysis, we found that inhibiting endocannabinoid receptors (CB1R) promoted synaptic cooperation by extending the time window in which thalamic synapses cooperate with cortical synapses[25]. To test whether inhibiting cannabinoid receptors also alters our memory cooperation time rule, we inhibited CB1R in the lateral amygdala, during the 1 h between events one and two (Supplementary Fig. 3A). According to our prediction, we found that inhibiting CB1R increased memory cooperation as seen by an increase in freezing to the CS- (Supplementary Fig. 3B, C) and a significant decrease in the discrimination index (Supplementary Fig. 3D). The promoting effect of inhibiting CB1R in synaptic cooperation was associated with a modulation of thalamic inputs by increasing the time windows in which thalamic inputs can cooperate with cortical inputs.

Having this in mind, we used chemogenetic manipulations of the thalamic and cortical inputs to the LA to assess their role in modulating the time rule of cooperation. Transfection of the Mgm thalamic nuclei with a vector expressing the hM3D(Gq) receptor resulted in the labeling of axons targeting the LA region via the internal commissure (Fig. 2A/A'). Using the 1-hour interval between events, in which memory association does not occur, we found that injection of CNO before the training trial, increased cooperation resulting in an increased freezing response to both CS- and CS + auditory cues (Fig. 2B, D). Analysis of the discrimination index shows that discrimination is significantly lower if thalamic inputs are activated by CNO injection in the presence of a vector expressing hM3D(Gq), compared to all other conditions (Fig. 2F). Interestingly, when plotting the level of freezing to the CS+ against the freezing response to the CS-, we found that chemogenetic activation of the thalamic inputs results in a significant correlation (Fig. 2G), a result similar to the 30-min interval. This result shows that activating thalamic inputs increases the time window of memory association, as we have observed at the synaptic level. We then asked whether activation of the cortical input also increases memory cooperation. To do this, we transfected the primary auditory region with the hM3D(Gq)

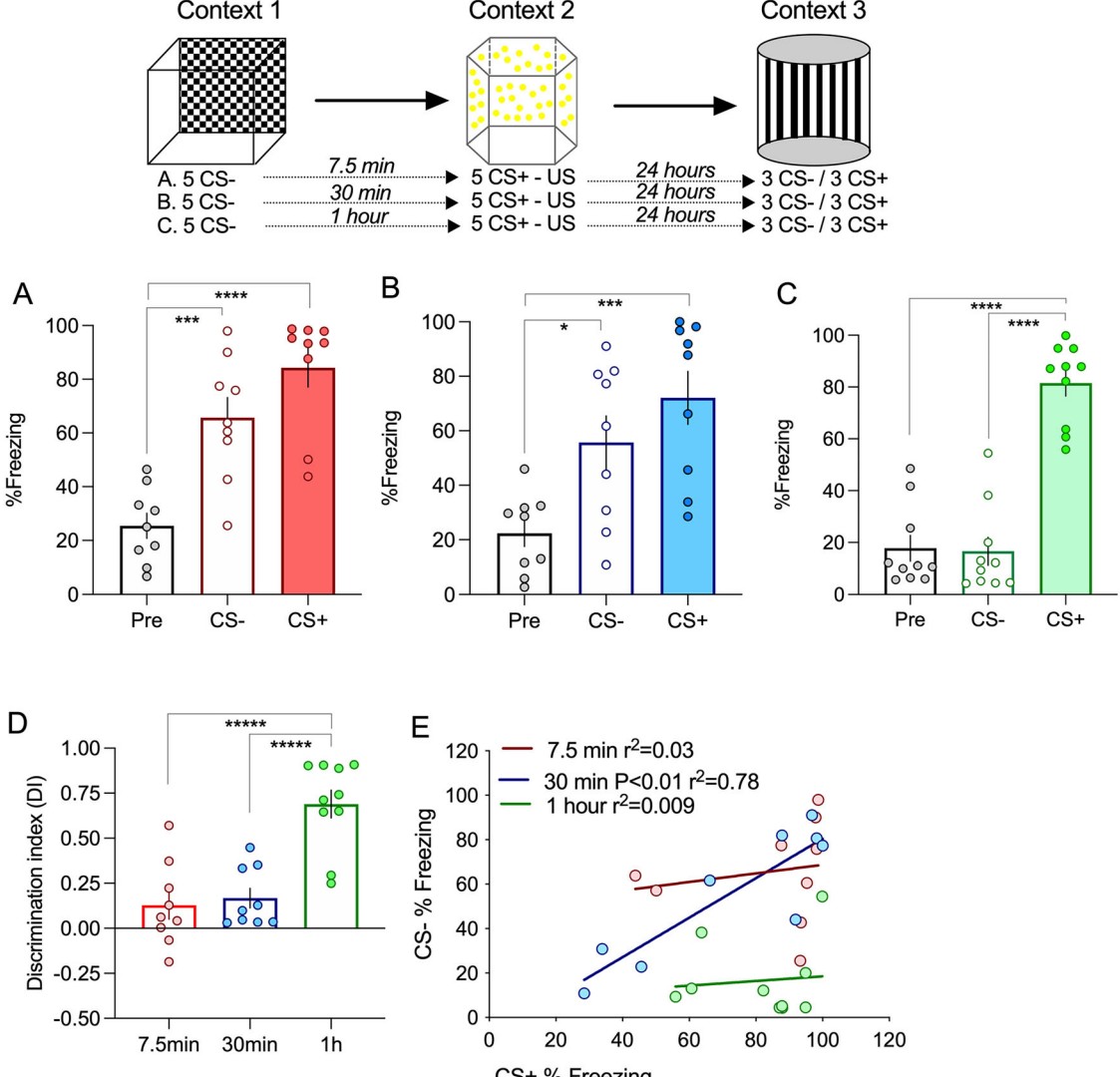

**Fig. 1 | Memory association depends on the time interval between events.** Diagram illustrates the sequence of trials during training (day 1) and in test (day 2). **A** When events are separated by 7.5 min, events are associated and animals significantly freeze above baseline responses to both CSs (ANOVA F = 20.87 $P < 0.0001$; ***$P = 0.0007$; ****$P < 0.00001$ $n = 9$). **B** Increasing the time interval between event 1 and 2 to 30 min also leads to fear memory association (ANOVA F = 8.98 $P = 0.0012$; *$P = 0.026$; ***$P = 0.001$ $n = 9$). **C** Increasing the time interval to one hour allows the dissociation between event 1 and 2, resulting in a specific fear response to the CS+ (ANOVA F = 53.47 $P < 0.0001$; ****$P < 0.0001$; ****$P < 0.0001$ $n = 10$). **D** Analysis of the discrimination index shows that only the interval of 1 h allows the dissociation between the neutral and fearful events (ANOVA F = 20.13 $P < 0.0001$; ****$P < 0.0001$; ****$P < 0.0001$). **E** Plotting the individual response to the two CSs reveals that for the intermediate time tested (30 min) there is a positive correlation between the fear response to the CS- and the CS+, correlation that is not observed in any other time interval tested. Error bars represent SEM.

receptor-expressing vector (Supplementary Fig. 4A). In this situation, the administration of CNO did not alter cooperation, and animals were able to discriminate between the two events, showing a preferential freezing response to the CS+ (Supplementary Fig. 4B, C).

**Memory competition is dependent on time and order of events**
In our synaptic study, competition was induced by the temporally related stimulation of three inputs to the LA (one cortical, two thalamic)[26]. Since synaptic competition results from an unbalanced distribution of PRPs among all activated synapses, the lower the availability of PRPs, the higher the competitive load. In our synaptic study, strong synaptic activation of the cortical input induced the synthesis of PRPs, whereas subsequent weak stimulation of two thalamic inputs created a competitive interaction. To mimic this scenario, we reversed the order in which memory association is induced by exposing the animals to the CS + US event before the CS-. Reverting the order did not change the behavioral outcome, with animals still associating the fearful and neutral events as before (Supplementary Fig. 5A).

Given that neurons in the MGm thalamic nuclei have a multimodal response, including visual stimuli, we added a third event in which animals are exposed to a third, different context and visual stimuli (green flashing LED). Again, based on our synaptic competition model, we predicted that competition would occur if the third event was temporally close to the two initial events, whereas increasing the elapsed time would leave intact the initial memory association. Competition was induced by exposing animals to the light trial 7.5 or 1 h after the CS+ and CS- events. In the absence of light, cooperation was observed (Supplementary Fig. 5A), whereas introducing the light at event 3 leads to a mild form of competition, reducing both responses to the CS+ and CS- (Supplementary Fig. 5B). As predicted, if the third event occurred 1 h after the initial CS + /CS- association, no competition was observed (Supplementary Fig. 5C). Given that the initial CS + /CS- association was done with the 30-min interval, in the absence of

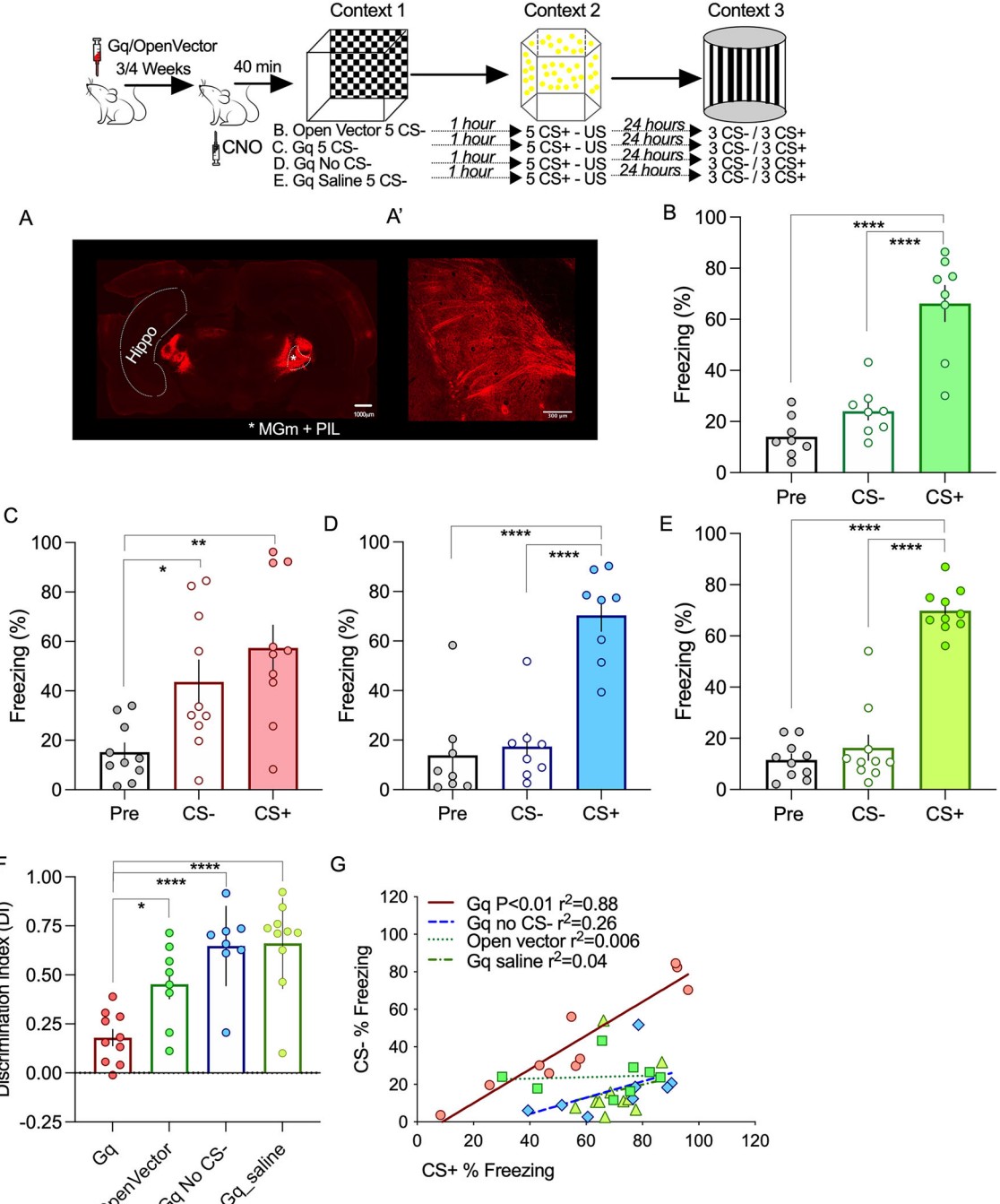

**Fig. 2 | Memory association is influenced by the activity of the thalamic inputs. A/ A'.** Injection of AAVs in the thalamic medial division of the medial geniculate nucleus (MGm and PIN) results in transfection of afferents that project to cortical areas and the lateral amygdala, by the internal capsule. **B** Using the one-hour interval, transfecting neurons with an open vector does not alter the ability of animals to distinguish between the neutral and aversive event (ANOVA F = 33.73 $P < 0.0001$; ****$P < 0.0001$; ****$P < 0.0001$ $n = 8$). **C** Increasing thalamic activity by CNO activation of the Gq DREADD led to memory association even if neutral and aversive event occur with an one hour interval (ANOVA F = 7.88 $P = 0.002$; *$P = 0.036$; **$P = 0.002$ $n = 10$). **D** Memory association is not due to the increase of thalamic activation per se since fear responses to the neutral event is dependent on

the presenting the CS- during training (ANOVA F = 25.74 $P < 0.0001$; ****$P < 0.0001$; ****$P < 0.0001$ $n = 7$). **E** Expressing Gq in MGm, in the absence of CNO activation is not sufficient to induce memory association (ANOVA F = 86.45 P < 0.0001; ****$P < 0.0001$; ****$P < 0.0001$ $n = 10$) **F.** Analysis of the discrimination index shows that increasing thalamic activity significantly promotes memory association (ANOVA F = 12.71 $P < 0.0001$*$P = 0.029$; ****$P < 0.0001$ ****$P < 0.0001$). **G** Responses to the neutral event (CS-) is correlated to the response of the aversive event (CS + ) if thalamic activity is increased. No correlation is observed in animals transfected with the open vector, in animals where the CS- is not present in training or in the absence of CNO. Error bars represent SEM.

the light stimulation, we observe that again freezing levels to the CS+ are correlated to CS- freezing responses (Supplementary Fig. 5D). Interestingly, although introducing the light resulted in the reduction of freezing responses, the correlation is still present, suggesting that the competitive

reduction of freezing responses was done to the CS + /CS- association as a block. In the situation where the light is presented 1 h after the initial CS + / CS- association, no change in the correlation was observed. We do not observe any significant freezing response to light in the test trial, suggesting

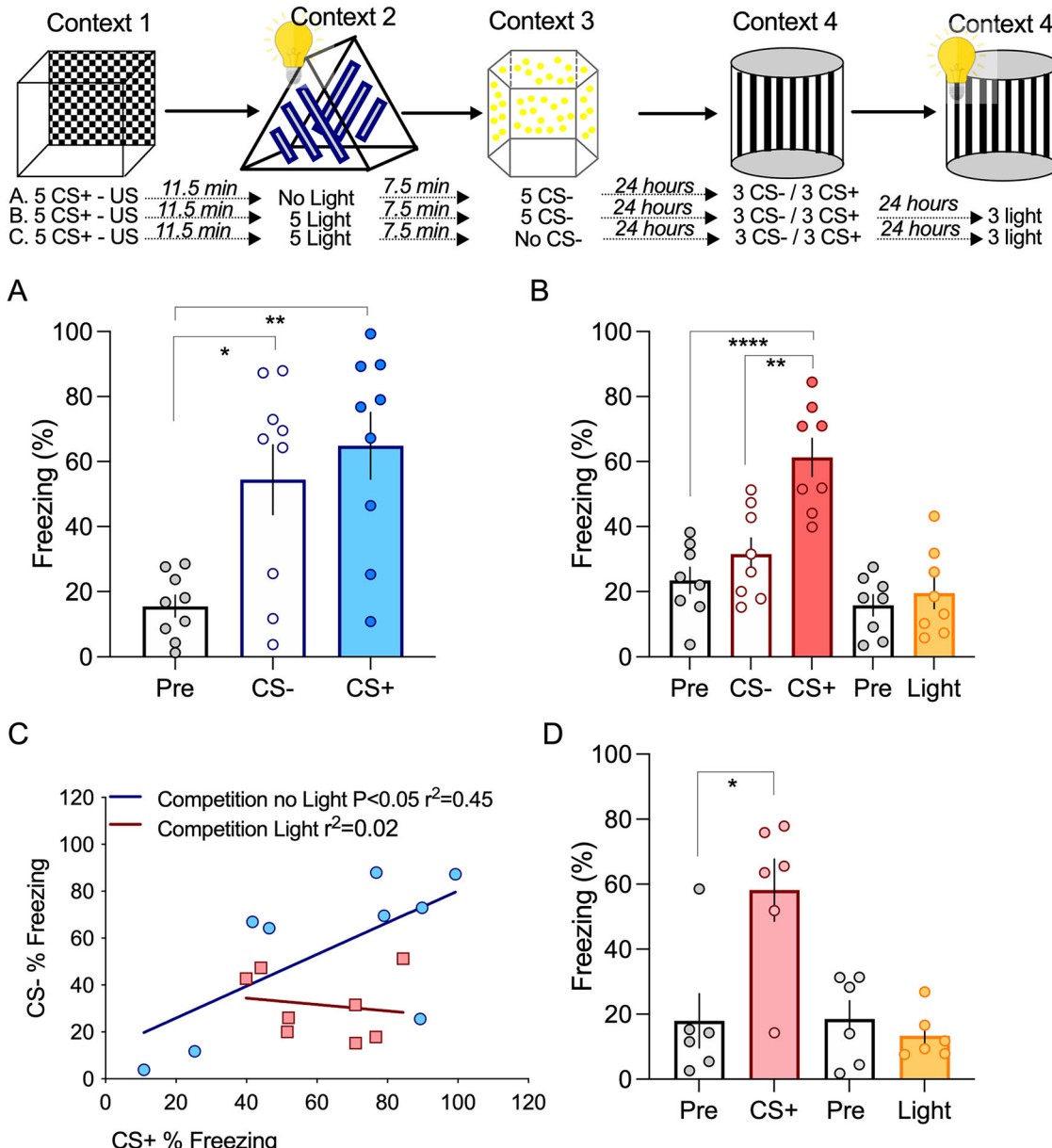

**Fig. 3 | Order of events determines competitive interactions between events.**
**A** Events are associated by cooperation if presented within a 30 min interval
(ANOVA F = 8.793 *P* = 0.0014; *\*P* = 0.012; \*\**P* = 0.0016 *n* = 9). **B** If animals are
exposed to a third event, consisting of a different context paired with a light stimulus,
between event 1 and 2, responses to the CS- significantly decrease (ANOVA F = 16.0
*P* < 0.0001; \*\*\*\**P* < 0.0001 \*\**P* = 0.0011 *n* = 8). **C** The light event can effectively
break the link between event 1 and 2, disrupting the correlation between fear
responses to the CS+ and CS-. **D** No association was seen between the fearful event
and the light event, even if the CS- event is not present (Unpaired two-tailed *t*-Test
pre-freezing vs CS+ \**p* = 0.01; Unpaired two-tailed *t*-Test pre-freezing vs light
*p* = 0.43 *n* = 6). Error bars represent SEM.

that memory association is not observed between multimodal stimuli in our
experimental conditions.

Given that competition by the light interfered with the freezing
response of both the CS+ and CS-, we designed a second competitive
experiment, in which the light event was present between the two sound
events (Fig. 3). Again, we used the 30-min interval between the CS+ and CS-
event and the light event occurred in between so that the time interval
between the light event and CS- event was 7.5 min. In this situation, we
observe that the light event effectively competes with the CS + /CS- asso-
ciation, resulting in a selective and significant decrease in the freezing
response to the CS- (Fig. 3A, B). This is also appreciated in the correlation
analysis (Fig. 3C). The light event disrupts the correlation between the CS+
and CS- freezing responses, again showing that the association is effectively
disrupted. As shown before, we do not observe any significant freezing

response to light in the test trial, even if the light event is present temporally
close to the CS + , suggesting that, in our experimental conditions, memory
association is not observed if events contain multimodal stimuli (Fig. 3D).

Given the multimodal properties of MGm neurons, we hypothesized
that the light stimuli, which induce competition, can be mimicked by
optogenetic stimulation of MGm axonal projections. To test this, we
expressed a channelrhodopsin in the MGm, which resulted in strong
labeling of axons in the internal commissure and the lateral amygdala
(Fig. 4A/A'). It is noteworthy to mention that, although the pattern of
expression of the viral vector is similar to the one observed for the
hM3D(Gq) DREADD, these manipulations have very different outcomes
regarding thalamic input activation. Excitatory DREADDs such as
hM3D(Gq) increase excitability to cues but do not trigger action potentials
directly[27]. Conversely, light activation of ChR2 expressing axons results in

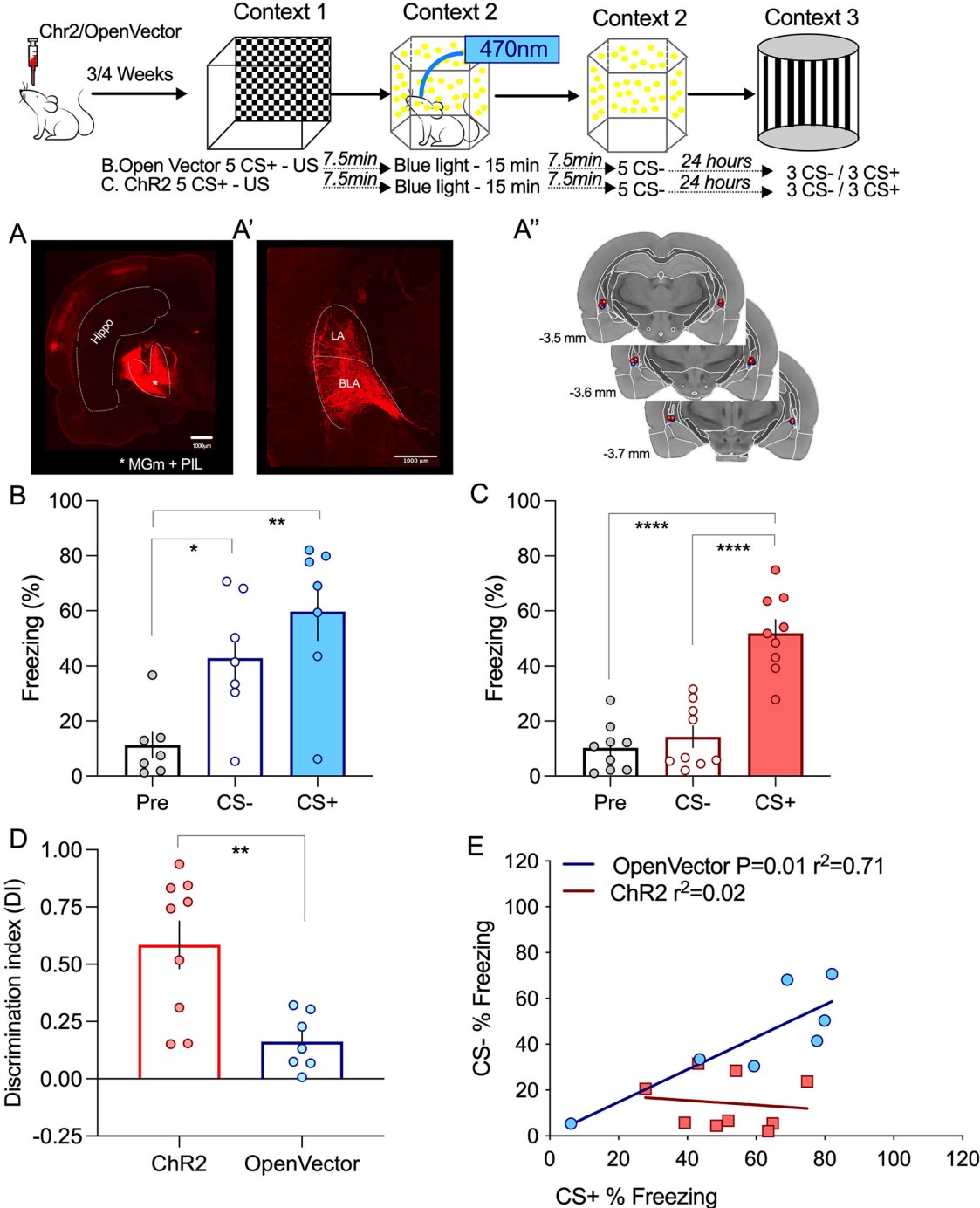

**Fig. 4 | Memory competition can be induced by direct activation of the thalamic input.** **A**/**A'**. Injection of AAVs in the thalamic medial division of the medial geniculate nucleus (MGm) results in transfection of afferents that project to cortical areas and the lateral amygdala, by the internal capsule. **A"** Location of the optical cannula used to deliver the light stimulation to thalamic inputs terminating in the lateral amygdala. **B** Using the 30 min interval, light-activation of thalamic transfected with an open vector does not alter the ability of animals to associate between the neutral and aversive event (ANOVA F = 9.001 $P$ = 0.002; *$P$ = 0.0352

**$P$ = 0.0016 $n$ = 7). **C** Increasing thalamic activity by light-stimulation of ChR2 transfected inputs significantly decrease the response to the CS- (ANOVA F = 34.17 $P$ < 0.0001; ****$P$ < 0.0001 ****$P$ < 0.0001 $n$ = 9). **D** Light stimulation significantly increased the discrimination index (Unpaired two-tailed $t$-Test **$p$ = 0.0044). **E** By inducing competition, light-stimulation of thalamic inputs effectively dissociates events 1 and 2 abolishing the correlation between CS+ and CS- responses observed in control conditions (open vector). Error bars represent SEM.

action potential triggering. This means that hM3D(Gq) DREADD expression modulates the response to the auditory pathways to the cues whereas ChR2 light activation triggers the activity of axons irrespective of whether they were recruited or not by the auditory cues.

Having this in mind, we observed that light stimulation of the amygdala in the time interval between the CS+ and CS-, in control conditions (open vector) resulted in memory association and significantly higher freezing levels for both stimuli, as seen before (Fig. 4B). Conversely, light stimulation in animals expressing channel rhodopsin in the thalamic inputs, led to competition, as seen by a significant decrease in the freezing response to the CS- (Fig. 4C). This is also seen in the analysis of the discriminative index, which is higher in channel rhodopsin-expressing animals (Fig. 4D).

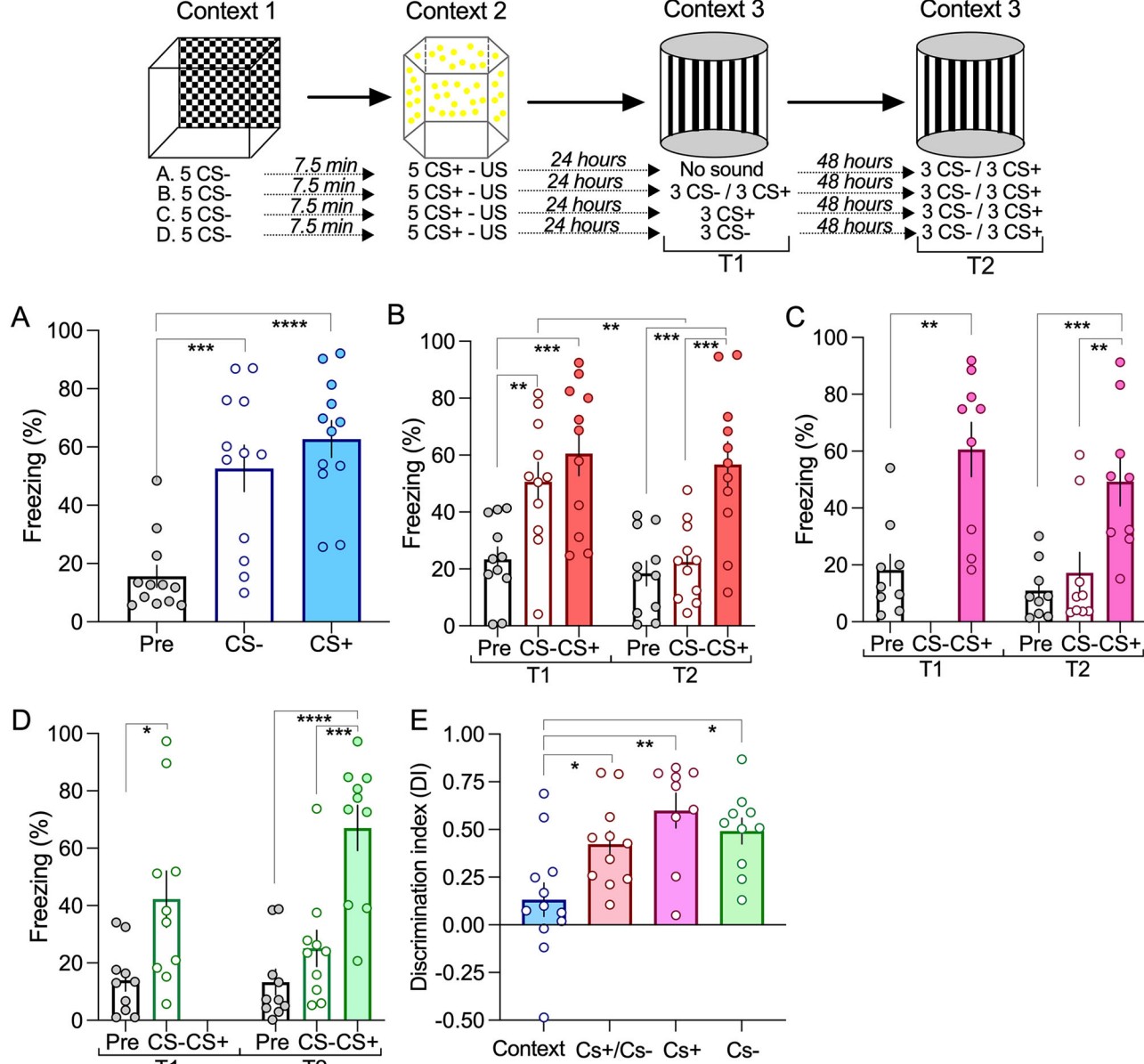

**Fig. 5 | Memory reactivation leads to a selective decrease in the response to the neutral CS- event. A** Exposing animals to a new context does not induce reactivation and animals respond to the CS+ and CS- in subsequent test event (ANOVA F = 15.86 $P < 0.0001$; ***$P = 0.005$ ****$P < 0.0001$ $n = 12$). **B** Presenting the CS- and CS+ in the first test event leads to a significant reduction to the CS- in test 2 (Two-way repeated measures ANOVA F = 7.69 $P = 0.0084$; Time1 **$P = 0.0072$ ***$P = 0.0002$ Time2 ***$P = 0.0001$ ***$P = 0.0006$ $n = 11$; Time 1 vs Time 2 CS- ** $P = 0.0024$). **C/D.**

Presenting the CS+ alone (**C**; Time1 Unpaired two-tailed $t$-Test **$p = 0.0014$; Time2 ANOVA F = 10.06 $P = 0.0007$ ***$P = 0.001$ **$P = 0.0047$ $n = 9$) or the CS- alone (**D**; Time1 Unpaired two-tailed $t$-Test *$p = 0.01$; Time2 ANOVA F = 19.7 $P < 0.0001$ ****$P < 0.0001$ ***$P = 0.0002$ $n = 10$) also resulted in a significant reduction to the CS- in test 2. **E** Memory reactivation significantly increase the discrimination between fearful and neutral events (ANOVA F = 6.53 $P = 0.0011$ *$P = 0.047$ **$P = 0.011$ *$P = 0.012$). Error bars represent SEM.

Interestingly, we found that light activation of thalamic inputs is sufficient to disrupt the correlation observed in control conditions, suggesting that activation of thalamic inputs efficiently modulates fear memory cooperation and competition (Fig. 4E).

### Memory reactivation leads to the loss of the CS- freezing response

One of the essential features of memory is its ability to update. In our cooperation design, the memory trace represents the association between the CS+ and CS- events. Given that memory update requires reactivation of the previous trace, we designed a reactivation experiment in which the previous memory is either updated by the presentation of both sound stimuli (CS+ and CS-), only one of the sounds (CS+ or CS-) and no sounds

(exposure to context only). Animals are then tested 48 h after this reactivation trial, by exposing animal to the CS+ and CS- (test trial). We found that if animals are only exposed to context, no change is observed at the test trial, showing that exposure to context is not able to disrupt the previously associated memory (Fig. 5A). In contrast, exposing animals to the auditory stimuli led to a significant decrease in the freezing response to the CS-, but not to the CS+ (Fig. 5B, C, D). This reduction in the response to the CS- in the test trial is observed irrespectively of whether animals are exposed to both CS+ and CS- or only to the CS- or to the CS +. This is also appreciated by analyzing the discrimination index in the test trial. Only the condition where animals were only exposed to the context showed a significantly lower discrimination reflecting a high freezing response to both stimuli, CS+ and CS- (Fig. 5E). These results show that reactivation is sufficient to trigger

memory updating leading to a decrease of the non-significant stimuli response.

Fear memories are generally enduring[28]. To test whether our cooperative memory fades with time, we tested whether testing animals one month after training translates into a decrease in the freezing response to the auditory stimuli. We observe that one month after training, animals maintain their response to both the CS+ and CS- (Supplementary Fig. 6A T1). As seen in the previous experiment, exposing animals to the auditory stimuli in the test trial led to a significant decrease in the CS- but not in the CS+ (Supplementary Fig. 6A T2). This is also evident in the significant increase in the discrimination index in T2 as compared to T1 (Supplementary Fig. 6B). This result shows that cooperative memories are enduring but very sensitive to disruption upon a single reactivation event.

## Discussion

The ability to link events, occurring in close temporal proximity, is highly relevant to acquiring associative memories. Equally important, or even more relevant, is the ability to separate events. Here, we addressed the temporal rules by which memories are associated or dissociated. Using a modified discriminative fear learning paradigm, we found that events occurring within short intervals, lower than 30 min result in the acquisition of an associative memory. If conversely, events occur separated by one hour they are not linked and thus, animals were able to discriminate between the fearful and the neutral stimuli, responding specifically to the fearful stimuli. Interestingly for the 30 min interval, we found that animals were distributed in two distinct groups, animals that respond by fearing both aversive and neutral stimuli and animals that do not respond to either. This observation suggests that for this intermediate time interval, in some animals, the association between neutral and aversive events interferes with the consolidation of a fear memory. These temporal rules mirror the heterosynaptic interactions between thalamic and cortical inputs into the lateral amygdala, previously described by us[25,26]. Importantly, this form of associative fear memory is not due to fear cue generalization nor to context generalization, given that if animals are not exposed to the CS- during training no response to this particular cue is observed in the test trial, even for the shortest time interval tested. Also, the acquisition of a fear response to the neutral auditory cue is dependent on the presence of the second auditory cue during the aversive event, showing that an association between the two auditory cues underlies the fear-associative memory.

Interestingly, as stated in the introduction, we previously found that thalamic synapses are only able to cooperate with the cortical synapses if activated within 7.5 min[25]. This time restriction of synaptic thalamic cooperation is due to the activation of cannabinoid receptors (CB1R)[25,26]. Consistently, we also found that fear memory association can be extended to 1 h if CB1Rs are inhibited in the amygdala. Although we have previously shown that presynaptic CB1Rs are expressed in the thalamic pre-synaptic terminals and that CB1R agonists have a profound effect on thalamic excitatory synaptic transmission[26], we cannot rule out a more generalized effect of CB1R antagonists, including modulation of inhibitory neurons and astrocytes that lead to an increase in arousal and thus fear expression[8,29,30]. We showed that increasing the activation of the thalamic input, by expressing an activating DREADD receptor increased memory cooperation leading to fear generalization. A similar manipulation of the cortical input did not alter the response of animals, which indicates that the increase in memory association is not due to an overall increase in amygdala activation. It is noteworthy to further discuss that increasing the thalamic input activity not only increases memory association but it leads to a response similar to the cooperation 30 min, in which the response to the CS- is tightly correlated to the response to the CS+. Given that the correlation emerges because there are animals in which the CS- interferes with the fear memory of the CS+, this observation suggests that increasing thalamic activation increases the time window during which events interact but does not determine the direction of such interaction. That is, increasing thalamic activity allows the association between the neutral and aversive events even if separated by one hour, but does not lead to an increase in the fear response

to both auditory cues in all animals. These results go in line with a previous study that showed a critical role of post-learning activity of the thalamic input in the consolidation of fear memories[31] and supports our hypothesis that thalamic activation determines whether animals associate or discriminate events.

Previous studies have shown that events can be associated in time[5,6]. Both these studies show that within a very large time window of 5 to 6 h events can be associated and evoke a fearful memory, contrasting with the time rules that we report. In the first study, by Cai et al, the authors use a form of context fear conditioning, in which contexts are associated before the pairing of one of these contexts with a foot shock[5]. Context fear conditioning is dependent on the recruitment of the hippocampal network[11] and thus different temporal rules may determine the linkage of hippocampal-dependent memories. Indeed, synaptic cooperation in hippocampal slices can occur in larger time intervals than the ones we described for the amygdala circuit[32,33]. In a second study, using auditory fear conditioning, the authors show a cooperative effect between events that relies on the association between two auditory cues similar to the one shown here[6]. However, the shortest interval tested by the authors was 90 min, which contrasts with the results presented here. This discrepancy might be due to the different model animals used in these studies. We have recently described that heterosynaptic cooperation between thalamic and cortical synapses follows different temporal rules in rats and mice[34], with a higher window of cooperation in mice. This is also consistent with our observation that thalamic inputs in rats express CB1R whereas in mice the expression is almost absent[26] and may explain the different time rules of fear memory association.

Our temporal rule also applies to memory competition. We found that exposing animals to a competitive event, after auditory fear memory association only destabilizes previous associative memory if occurring within a short time interval. The competitive effect is mild and it is important to note that in our experiment setting the light neutral event was never associated with the aversive event. These results suggest that in our experimental design, we do not find a multimodal memory association (CS + - light association) which might explain the mild competitive effect observed. Importantly, when the competitive event occurs between the two auditory events, we observe that memory linkage is abolished, shown by eliminating the correlation between the CS- and CS+ response. As for cooperation, optogenetic activation of the thalamic input is sufficient to mimic the light competition, disrupting the association between the neutral and the fearful event. The effect of activating thalamic input is different whether one is in a cooperative or competitive scenario. If the two events are separated by 30 min we are in a cooperative scenario and in this case, further activation of the thalamic input favors competition, increasing discrimination towards the fearful cue. In the 60 min interval, in control conditions, there is no cooperation and thus, activation of the thalamic input increases the time window during which the two events can be associated.

Although enduring, the fear-associative memory for the neutral event is very sensitive to reactivation. Re-exposing animals to any of the auditory stimuli, leads to the loss of the response to the CS-, showing that reactivation promotes fear discrimination. Our results resonate with a previous study showing that a distractor at the time of memory reactivation can induce a competitive disruption of a previously acquired fear memory[35]. This suggests that once reactivated, memory is updated towards maintaining the most relevant event. Taken together, we show a clear time rule determining whether events are associated by cooperation or interfere by competition in fear memory acquisition and maintenance. We also show that the modulation of thalamic synaptic activity has a critical role in determining this temporal rule. In our experimental conditions, the acquisition of the cooperative fear memory relies on the association between the two auditory cues and thus, further characterization of the underlying mechanisms is necessary to extrapolate our temporal rule to other forms of learning. Given the individual differences in memory association and the proposed role of the endocannabinoid signaling system in its modulation, we propose that individual differences in the endocannabinoid system may explain

individual susceptibilities to fear generalization by erroneous associations between neutral and fearful events.

## Materials and methods

### Animals

Naïve male Sprague-Dawley rats (200–350gr) obtained from a commercial supplier (Charles River Laboratories, France) or an in-house breeding stock at NOVA Medical School animal facility, were used in this study. Animals were allowed to acclimate after transportation for at least one week after arrival. Rats were housed in a temperature- and humidity-controlled housing facility under a 12 h light-dark cycle (lights ON at 8 a.m. OFF at 7 p.m.), with *ad libidum* access to food and water. All the behavioral procedures were performed during the light phase of the cycle. Animals were handled for two days before each experiment. Rats were housed in two per cage, except between training and the first testing session, in which animals were housed alone. In the experiments where testing was performed with 30 days interval from training, animals were housed alone for 24 h (after training) and then in pairs for the remaining duration of the behavioral trial. All procedures that involved the use of rats were approved by the Portuguese Veterinary Office (Direcção Geral de Veterinária e Alimentação — DGAV) and the ethics committee of NOVA Medical School.

### Fear conditioning paradigm

All the behavioral studies were performed in sound-attenuating boxes present in the same procedure room. Four different conditioning boxes were used for conditioning and testing sessions. To minimize generalization between contexts, the boxes differ in size, shape, color, odor, lighting conditions, and floor composition. One context consisted of black acrylic walls ornamented with a chess pattern (black and white) and cleaned with 1% lavender detergent. The second context, was made of clear Plexiglass walls, ornamented with yellow circles, and cleaned with 1% acetic acid. The third context, consisted of black acrylic walls, ornamented with blue/white stripes, and cleaned with 1% marine detergent. The forth context consisted of clear Plexiglass walls, ornamented with vertical black stripes, and cleaned with 70% ethanol. Photos of conditioning boxes can be found in supplementary data. All the chambers contained an electrical grid floor (that is covered during the CS- presentation and probe test); a speaker through which auditory stimuli were delivered, a house light, a fan, and an infra-red light. The different chambers were used in a counterbalanced manner across experiments; therefore, the numerical labels assigned to the contexts in the figures do not correspond exactly to the order in which the boxes are described in this section.

### Memory cooperation experiments

Animals were exposed to two tones, one paired with a foot shock (CS + : continuous auditory tone at 2 kHz, 10 ms rise and fall, 75 dB, 20 s) and one unpaired (CS-: auditory pips, 10 kHz pips repeated at 0.5 Hz, 10 ms rise and fall, 75 dB, 20 s). Unless stated, the two auditory cues were not used counterbalanced. Rats were placed in a conditioning chamber and allowed to explore for 2 min before any auditory tone was presented. After this, animals were exposed to a block of five presentations of the CS- (in a pseudo-random manner, iTi 60–180 s). After different time intervals –7.5 min, 30 min, or 1 h – animals were transferred to another context and conditioned with a block of five presentations of CS+ that co-terminated with a foot shock (1 mA, 1 s). After training, rats remained in the chamber for an additional 30 s before returning to their home cage. On day 2, animals were tested (probe test) in a novel chamber by presenting three trials of the CS- followed by three trials of the CS+ (same settings as in training, pseudo-random presentation, iTi 60-180 s). In the case of the reactivation and remote experiments, animals were tested twice, on consecutive days. In that case, the same context was used for testing (T1 and T2).

### Memory competition experiments

In memory competition experiments, we used a light stimulus (Green LED, Intensity: 35 LUX, Duration: 0.48 s; Interval: 0.86 s) as a CS3. As before, we used two auditory stimuli, CS- and CS + , with the same parameters as in cooperation experiments. In this experimental setting, the CS+ was presented first, after which the animal transitioned to another context where it remained until the second event was initiated (five presentations of the CS-; see diagram in figures). The light stimulus was presented in a third context either after the CS- presentation (first experiment) or in between the CS+ and CS- events. The total time interval between CS+ and CS- events was always 30 min. On day 2, animals were tested for CS- and CS+ stimuli, as described for cooperation experiments. The light probe test occurred on day 3, presenting three trials of the light stimuli in the same context as in probe test 1.

### Viruses

Adeno-associated viral vectors (AAV) were purchased from Addgene. For the optogenetic experiments, we used serotype 1, pAAV-CaMKIIa-hChR2(H134R)-mCherry, a gift from Karl Deisseroth (Addgene plasmid#26975; http://n2t.net/addgene:26975;RRID:Addgene_26975), titer $1 \times 10^{13}$ vg/mL and the serotype 1, pAAV-CaMKIIa-mCherry a gift from Karl Deisseroth (Addgene plasmid#114469; http://n2t.net/addgene:114469;RRID:Addgene_114469), titer $1 \times 10^{13}$ vg/mL. For the chemogenetic experiments we used the serotype 9, pAAV-CaMKIIa-hM3D(Gq)-mCherry a gift from Bryan Roth (Addgene plasmid #50476; http://n2t.net/addgene:50476;RRID:Addgene_50476), titer $1 \times 10^{13}$ vg/mL and serotype 9, pAAV-CaMKIIa-mCherry a gift from Karl Deisseroth (Addgene plasmid#114469; http://n2t.net/addgene:114469;RRID:Addgene_114469), titer $1 \times 10^{13}$ vg/mL.

### Surgery and viral transfection

Rats, 6–8 weeks of age, were anesthetized with isoflurane (4%; (SomnoSuite anesthesia system, Kent Scientific) and maintained at 1.5–2% throughout the surgery in the stereotaxic setup (Kopf 940, Kopf Instruments). Surgery started with a skin incision, after local injection of 2% lidocaine, and retraction to expose the skull, where a craniotomy and durotomy were performed. AAVs were delivered bilaterally into the target brain areas using a pulled glass pipet (tip diameter 20–30 um) connected to a manual injector (WPI instruments). A volume of 0.3uL (for A1) or 1uL (for MGm), per hemisphere, was injected, over 10 min. After injecting the virus, the needle was left at the site of injection for an additional 10 min to allow the virus to diffuse into the tissue. After that, the skin was sutured. Coordinates (from bregma) for targeting the different brain areas were the following: auditory cortex (A1): -3.6 (AP); ± 6.4 (ML); -5.0 (DV from the skull surface); MGm: -5.28 (AP); ± 3.40 (ML); -6.20 (DV from the skull surface), LA: -3.00 (AP); ± 5.20 (ML); -7.20 (DV from the skull surface); all in mm, according to Paxinos and Watson (1986). In the case where implants were needed (optogenetics and in vivo pharmacology), the implants were fixed to the skull with anchored screws, medical glue (Vetbond), and dental cement (Dentalon Plus, Kulzer). Optic fiber cannulas were lowered to 7.2DV below the skull surface. Thirty min before the end of the surgery, animals were administered carprofen (5 mg/Kg; repeated at day 2 post-surgery) and 1 mL of saline solution for hydration. Animals were then placed in a heated cage post-surgery for recovery. For optogenetic and chemogenetic manipulations, rats had 4-5 weeks of recovery to allow adequate virus expression.

### In-vivo pharmacology

For in vivo pharmacology, the experiments were done 1 week after the guide cannula implantation. Guide cannulas were implanted over the amygdala, using the same AP and ML coordinates as described previously. Rats were implanted bilaterally with a 7.2 mm guide cannula (26 gauge; PlasticsOne, Bilaney). The guide cannula was fixed to the skull using dental acrylic and two bone screws. Dorsoventral coordinates were measured from the skull surface with the internal cannula (33 gauge; PlasticsOne, Bilaney) extending 0.5 mm beyond the end of the guide cannula. Intracranial infusion was

administered using an injection needle (Plastic One, USA) inserted through the guide cannula with the injection needle connected to 0.5 µL syringes with polyethylene tubes and controlled by an automated microinjection pump (Precision Pump, Longer). AM281 solution (25 µM, HelloBio), diluted in 1% of DMSO, of a volume of 200 nL was injected 5 min after the end of CS- presentation at a rate of 20 nL/min. The control animals were infused with only vehicle solutions.

### Chemogenetics
Adeno-associated viral vectors expressing the hM3D(Gq) receptor (described above) were injected either in the MGm thalamic region or in the primary auditory cortex, both areas projecting to the LA region of the amygdala. A 3-4 weeks expression time was given to allow a stable expression in the axons. After this period, for three days, rats were habituated to intraperitoneal (IP) injections of saline before CNO (4 mg/Kg, HelloBio) was administered, 40 min before the training session. After each experiment, the verification of the transfection area was performed by PFA brain fixation and cryostat brain slicing.

### Optogenetics
Channel rhodopsin expressing AAV and open vectors (described above) were injected in the MGm thalamic region projecting to the LA, and a 3-4 weeks expression time was given to allow a stable expression in the axons. Two 200-micrometer (Thorlabs Fiber Optic Cannula, 2.5 mm OD Ceramic, 200um, 0.39NA, L = 10 mm) optic fibers were implanted in the same surgery to target the LA. Fibers were inserted and glued to a cannula guide (Thorlabs OGF-5) and lowered to 7.1 mm below the skull surface. To stimulate ChR2, a 470 nm high-power LED light source was used with a light intensity of 7 mW/mm2. The end of each optical fiber was connected to the LED light source (470 nm, BLS-LCS-0470 Mightex Systems, Toronto, Canada), controlled by an LED driver (BLS-SA04-US Mightex Systems) and a computer. Transfected terminals in the amygdala are light-stimulated with 30 pulses at 0.033 Hz, 75% of LED intensity, delivered bilaterally, during the interval between the CS+ and CS-. Optogenetic stimulation occurred for 15 min, starting 7.5 min after the transition to context 2 and was terminated 7.5 min before CS- presentation. All the freely moving experiments were done with a rotary joint (Thorlabs). After each experiment, the verification of the optic fiber tract and area of transfection was performed by PFA brain fixation and cryostat brain slicing.

### Imaging
At the end of the experiments, animals received an overdose of isoflurane and the brains were removed from the skull and fixed in buffered 4% paraformaldehyde (pH = 7.5) overnight. Brains were then sectioned with a cryostat (Leica CM3050 S) in sections of 40 µm thickness. Imaging was performed using a BC 43 bench confocal microscope (Andor, Oxford Instruments). Whole brain slide tile images were taken using a 2.5X (0,06) or 20X objective (0,8). The emission wavelength for mCherry was 590 nm with 650 ms of exposure time. Brain slices were inspected to confirm the virus expression in the thalamic and cortical regions projecting to the LA, as well as expression within the amygdala areas. Brightfield images were taken to determine the cannula or optic fiber location in the LA. Images of the expression pattern of all animals included in this manuscript can be found in the supplementary data. Representative images do not represent the site of injection but rather projection patterns of transfected neurons.

### Data collection and statistical analysis
Animal behavior was recorded using a video camera. A semi-automated video tracking software (Bonsai) was used to quantify behavioral scores. Behavioral performance was measured based on the percentage of freezing to baseline responses (pre-freezing). Freezing was scored from the video recording using a frame-by-frame analysis of pixel changes. Freezing responses evoked by the three auditory stimuli were averaged. Pre-freezing corresponds to the average freezing of the 120 seconds before tone presentation. Animals that showed a pre-freezing response above the mean+SD of the group were excluded. Fear discrimination index (DI) was calculated by DI = [(CS+ ) -(CS-)]/[(CS+ ) + (CS-)]. Statistical analyses were done with Prism 10 (GraphPad Software, San Diego, CA, USA). In group analysis, data are presented as mean ± SEM, and the number of animals in each group is depicted in the figure legends and the supplementary statistical table 1. Before choosing the statistical test, a normality test (Shapiro-Wilk normality test) was done on all data sets. Given that all our data sets presented a normal distribution, then parametric tests were used. When comparing three or more groups a one-way ANOVA with a Tukey's post hoc test for multiple comparisons was used. When comparing two groups, a Student t-test was used. In the case of the reactivation and remote experiments, given that animals are tested two times, a repeated-measures ANOVA was used with post-hoc correction of significant interactions. Main effects, $P$ values for ANOVA, and Post-hoc multiple comparisons, as well as $t$-test $p$-values and effect sizes, are mentioned in the figure legends and summarized in the supplementary statistical table 1. For the correlation analysis, the percentage freezing to the CS+ was plotted against the percentage freezing to the CS-. A Pearson correlation analysis was performed to assess the statistical significance of correlations. R-squares and $p$-values are depicted in the figure legends and the supplementary statistical table 1.

### Reporting summary
Further information on research design is available in the Nature Portfolio Reporting Summary linked to this article.

### Data availability
Data used in all figures and analysis in this manuscript is available as supplementary data 1 and can also be found here doi: 10.5281/zenodo.15412058. Raw source data will be available upon request to the corresponding author.

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

## Acknowledgements
We would like to thank Ana Carolina Temporão for developing the scripts used to run the behavioral experiments. We would also like to acknowledge the support of the Imaging facility at i3S and the Animal Facility at i3S. This work was funded by National Funds through FCT—Fundação para a Ciência e a Tecnologia, I.P., under the project 2022.09002.PTDC and a Grant from the Brain and Behaviour Foundation to RF (NARSADYI-2016-GA25118). RF was supported by an CEEC individual grant (2020.02221.CEECIND; DOI 10.54499/2020.02221.CEECIND/CP1586/CT0013), NM was supported by a FCT PhD fellowship (SFRH/BD/130911) and IC is supported by a research grant in the scope of the Fundação para a Ciência e Tecnologia (FCT) Project Grant (2022.09002.PTDC, DOI 10.54499/2022.09002.PTDC).

## Author contributions
Conceptualization: R.F.; Data Curation: R.F., N.M., I.C.; Data Analysis: R.F., N.M., I.C.; Funding acquisition and Project administration: R.F.; Supervision: R.F.; Writing – original draft: R.F., N.M.; Writing – review & editing: R.F.

## Competing interests
The authors declare no competing interests.
