## [Transparent Peer Review file · Communications Biology]

Thalamic input to the lateral amygdala determines the temporal window of fear-memory association

Corresponding Author: Dr Rosalina Fonseca

Version 0:

Reviewer comments:

Reviewer #2

(Remarks to the Author)

In this manuscript, the investigators provide evidence that neutral and fearful events become associated when both occur within a 30-min window of time. They also show that introduction of a third learning event results in memory competition. Through optogenetic and chemogenetic manipulations, the investigators demonstrate that the neutral/fearful event association depends on activation of the MGm inputs to the lateral amygdala.

Overall, the manuscript is well-written. The area of research would be of interest to those researchers who study the neural mechanisms underlying learning (and more specifically, learning that involves fear). Although the authors propose that their data suggest a linkage of neutral and fearful events, I am not convinced that the data strongly support this conclusion. I have several concerns / recommendations that are noted below:

- 1) In the introduction, the authors need to more clearly differentiate between cooperative and competitive interactions, particularly at the synaptic level. This includes providing a more specific description of cooperative maintenance of LTP.
- 2) The acronym PRP should be defined when it is first used (on p. 2 instead of p. 9).
- 3) The research includes males only. Why were females not included? This is a major limitation regarding generalizability of the findings.
- 4) The differences between the CS+ and CS- could be clearer. It is not clear why two auditory stimuli were used for the CS+ and CS-; it is also not clear why the third stimulus was from a completely separate sensory modality (i.e., visual).
- 5) All of the figure insets need to be more clearly labeled. As is, the reader must read through the legend to figure out what each inset is depicting. It would be much more helpful to have them labeled (e.g., Fig 1A is labeled "7.5 min", Fig 1B is labeled "30 min", and so on...).
- 6) The authors suggest that "The cooperative effect seen in the 7.5 min interval is not due to an overall generalization of fear...". The effect may not be due to an "overall" generalization of fear, but it most certainly could be due to a generalization between two different auditory stimuli. How can the authors rule out the possibility that the rats are simply demonstrating more fear to one auditory stimulus (CS-) because it was presented close in time to another auditory stimulus (CS+) that was paired with shock?
- 7) Throughout the manuscript, the rats' fear responses to the CS- are quite variable. Indeed, in each case, there appears to be a bimodal distribution (see Figs. 1A, 1B, 2C, 3A, 5A). The highly variable nature of the freezing in response to the CS- weakens the argument that the neutral and fearful events are "linked" in all animals. It would be interesting to know what factors, at the neural level for instance, can explain such variability.
- 8) On p. 6, the authors state "One group of animals freezes to both stimuli ... whereas a second group does not freeze to either stimulus." The second group does freeze, just less. This needs reworded.
- 9) On p. 6, the authors suggest that animals in the 30-min interval "generalize" their responses, choosing to either generalize

freezing or not freezing. In the previous paragraph, the authors argue that the association effect is not due to an overall generalization of fear. The jargon in paragraphs one and two can get mixed. This needs more careful attention to wording. Moreover, the relationship between CS+ and CS- freezing is correlational; no causal inferences can be made here.

10) It is unclear what novel information is being presented by the results displayed in Fig. 3. The inclusion of a third event (with a light stimulus) prevents the generalization made between the neutral and fearful events. However, this seems like a simple case of memory interference. Plus, the added stimulus is from a different sensory modality.

11) It would be useful to know if the response to the CS- is different from the response to the CS+ throughout the experiments. In some cases, fear to the CS- appears to be significantly less than fear to the CS+, even though it is greater than "pre".

12) In Fig. 5D, it would be useful to know if CS- under T1 is different from CS- under T2.

Reviewer #3

(Remarks to the Author)

This manuscript by Madeira and colleagues investigates the role of thalamic MGm inputs, as opposed to cortical inputs, to the lateral amygdala in regulating the memory association between temporally proximal events. This work is novel and provides new insights into the mechanisms underlying memory linking. Although the findings are specific to this circuit and memory type, they are of significant interest to the broader memory research community, especially in light of ongoing efforts to uncover the molecular and circuit-level mechanisms that enable memory associations.

The interest in this study stems from the limited understanding of memory linking and the variability in findings about the timing of memory associations, which appear to depend on, as this paper and prior work by the authors suggest, task design, brain regions involved, and molecular engagement.

I found the paper to be innovative and suitable for publication. However, I have some comments and suggestions to enhance the manuscript's clarity and address certain points:

Major Comments

1. The authors discuss memory generalization, but an alternative explanation could be that some mice generally fail to form memories when two events are spaced 30 minutes apart. To address this, I suggest conducting an additional test 24 hours or more after the 30-minute interval test to verify whether the memory persists in animals that exhibit reduced freezing. This phenomenon should also be discussed in greater detail, considering the possibility that presenting two events with a 30-minute interval might impair memory formation itself (resulting in reduced freezing in both contexts). The discussion briefly mentions this possibility ("in some animals, the neutral event can interfere with the fear memory acquisition of the aversive event"), but the mechanisms and alignment with the proposed hypothesis need further elaboration. Understanding this baseline phenomenon without manipulations is crucial for interpreting the circuits and molecular mechanisms involved and trying to reconciling the differing timings reported in previous studies.

2. The sentence "activating thalamic inputs reduces the time window of memory cooperation" seems to contradict the study's findings. Please clarify this point.

3. When CB1 was inhibited during a 1-hour delay, the authors observed increased memory association. However, no mice exhibited reduced freezing in these conditions as seen in the 30-minute control experiments. Could this reflect heightened anxiety? Although the discussion addresses this possibility briefly, I recommend performing anxiety tests to isolate the drug's specific effects.

4. The authors state that "thalamic activation increases the time window... but does not determine the direction of such interaction," but later claim that their results support "the hypothesis that thalamic activation determines whether animals associate or discriminate events." These statements seem contradictory and require rephrasing for consistency.

5. The authors claim consistency between their findings and the 2016 Rashid study, but Rashid used a minimum interval of 90 minutes. This discrepancy makes the comparison unclear, and I recommend rephrasing this discussion for clarity.

Minor Comments

1. Page 6: Figure 1B is not cited in the text.

2. Figure Titles: Adding titles to individual panels in all figures would improve readability (e.g., for Figure 1, titles such as "7.5 min interval," "30 min interval," and "1h interval" for panels A, B, and C, respectively).

3. Figure 2A Annotations: References or labels for the targeted brain regions visible in the figure would be helpful.

4. Color Coding: The same three colors are used to represent both contexts and intervals, which can cause confusion. I suggest adopting a different color scheme.

5. Figure 3B Clarity:

The distinction between "pre" and "pre group" is unclear.

Specify when the light group is analyzed (e.g., during light exposure or during the CS+ phase).

Reviewer #4

(Remarks to the Author)

Madeira et al showed that the temporal window between auditory CS- and CS+ determines if CR is associated with CS-. At 7.5-min or 30-min, not 1-h gap between CS- and CS+, rats showed freezing to CS-. They transfected MGm and PIN and showed that chemogenetic inhibition via systemic CNO led to CS- freezing at the 1-h condition. They showed light in between CS+ and CS- led to disruption of the CS- association, likely due to competition, which effect was also seen with optogenetic stimulation of thalamus-LA terminals.

The authors reported interesting observations that will need more information and control studies to convincingly support their conclusions.

Given that transfection can be transsynaptic, the authors need to rule out this possibility to support the main conclusions on thalamic input to the lateral amygdala.

Appropriate control groups are needed. In Fig. 2, Gq with vehicle control is missing. Gq with CNO in the lateral amygdala is also missing, which would be critical to support the sufficiency for this pathway in CS- freezing.

In Fig. 3, the hypothesis for the light stimuli in the competitive study can be arbitrary and more data reporting and control studies will be needed to make the competition concept convincing. How to rule out the possibility that CS- was associated with the recent event with light which was safe? It requires a control group with context 2 and no light in between CS+ and CS- to show that this effect is specific to light and not due to a novel and safe context. The freezing to CS- and to the context 30min after conditioning should be provided for the group in Fig. 3A and freezing to light, CS-, and contexts soon after conditioning should be provided for the group in Fig. 3B. These will enable understanding if association or generalization had already happened soon after conditioning. As Fig. 3D showed that Light did not lead to significant freezing on the next day, why is it not associative like the CS- and how did it lead to 'competition' when itself was not sufficient as a potential associative stimulus?

In Fig. 4, the extent of transfection in LA should be made clear in relation to fiber optic cannula tips in A'/A". The specific time point of optogenetic stimulation between CS+ and CS- should be clearly described. Since optogenetic stimulation is very transient and activities would likely return to normal during CS-, how did this lead to observed disruption? Brain regions, contours, and scale bars need to be superimposed in all histology figures to enable visualization of the transfection areas. A schematic image is needed to summaries variation in transfected areas across rats.

More information is needed to enable unbiased conclusions and reproduction. For example, are all contexts novel and counterbalanced? How are contexts A-D matched to contexts 1-2, 1-3 or 1-4? Photos of contexts A-D should be provided. It seems that the auditory stimuli used as CS- and CS+ were not counterbalanced. Would this affect the generality of the observed phenomenon?

To show that the CS- freezing reflects associative learning, more evidence beyond the CS- omission control in sup. Fig. 1 will be needed.

The procedure diagrams on top of each figure should make it clear as to how many groups were done and described in data figures below (as in Fig. 5).

Version 1:

Reviewer comments:

Reviewer #2

(Remarks to the Author)

The authors have satisfactorily addressed each of my concerns.

Reviewer #3

(Remarks to the Author)

The authors have addressed all of my previous comments in a thorough and satisfactory manner. I have no further comments and support the manuscript in its current form.

Reviewer #4

(Remarks to the Author)

The authors have addressed the major comments that I raised by providing more data of control experiments or by clarifying the texts. The photos of the apparatus should be included in the methods to aid reproducibility. There seems to be inconsistency in the texts (and photos) and in schematic drawings as to what geometric shapes of conditioning or testing contexts were used. This can certainly have implication on shape-related generalization. The expression patterns also varied a lot in the images provided (some outside of the MG), so the range of expression areas should be provided for the readers to see in a clear way. The authors should also check for consistency and accuracy in writing and in figures (e.g.

saline or DMSO in Supplementary figure 2, 1 month or 1 day after training and missing groups C and D in Supplementary figure 6).

We would like to thank the reviewers for all their clear and constructive comments. Please find below a detailed response to the questions raised.

Detailed response to reviewers

Reviewer #1

1) In the introduction, the authors need to more clearly differentiate between cooperative and competitive interactions, particularly at the synaptic level. This includes providing a more specific description of cooperative maintenance of LTP.

Thank you for the suggestion. We have revised the manuscript text accordingly.

2) The acronym PRP should be defined when it is first used (on p. 2 instead of p. 9).

We apologize for this mistake. We have revised the manuscript text accordingly.

3) The research includes males only. Why were females not included? This is a major limitation regarding generalizability of the findings.

Totally agree. We will make this point clearer in the discussion. The initial studies from which this work derived (Fonseca, 2013; Madeira 2020) used only slices from male animals and therefore we chose to maintain this selection for the behavioral study. Currently, in a follow-up project, we are including female animals to specifically clarify this point. However, due to several limitations, we are not able to respond to this question within the timeline of this work.

4) The differences between the CS+ and CS- could be clearer. It is not clear why two auditory stimuli were used for the CS+ and CS-; it is also not clear why the third stimulus was from a completely separate sensory modality (i.e., visual).

Thank you for the suggestion, we have revised the text to better clarify the differences between the CS- and CS+. We used two auditory stimuli since we are targeting auditory areas in our chemogenetic and optogenetic experiments. Additionally, there is a wide and strong background knowledge regarding auditory fear conditioning, including frequency responses, learning paradigms, and circuit analysis relating behavior to circuit plasticity. Interestingly, electrophysiological characterization of MGm neurons (thalamic) shows that these cells are multisensorial whereas MGv cells are not. Thus, using a stimulus from a different modality, in this case, a light, we aimed to more selectively stimulate the MGm. We revised the text to clarify this point.

5) All of the figure insets need to be more clearly labeled. As is, the reader must read through the legend to figure out what each inset is depicting. It would be much more helpful to have them labeled (e.g., Fig 1A is labeled "7.5 min", Fig 1B is labeled "30 min", and so on...).

We thank you for the suggestion. All figures have been revised to clarify differences in protocols used to facilitate matching between different protocols and the results.

6) The authors suggest that "The cooperative effect seen in the 7.5 min interval is not due to an overall

generalization of fear...". The effect may not be due to an "overall" generalization of fear, but it most certainly could be due to a generalization between two different auditory stimuli. How can the authors rule out the possibility that the rats are simply demonstrating more fear to one auditory stimulus (CS-) because it was presented close in time to another auditory stimulus (CS+) that was paired with shock?

Thank you for raising this point. Indeed, we conclude that animals are responding more to the CS- because it was presented close in time with a second auditory stimulus. However, this association is not generalization given that generalization refers to a response evoked by a stimulus that is not present during the training session. In the experiment depicted in supplementary figure 1A, the CS- is not presented during the exposure to context 1 and thus does not evoke a fear response in the test session.

We have performed an additional experiment to directly show that the association is done between the two auditory stimuli and not between the first cue (CS-) and the shock itself. In the new experiment, depicted in supplementary figure 1B, animals are exposed to the first context where the CS- is presented and 7.5 minutes later move to a different context and receive 5 foot-shock without the CS+. In this situation, in the test session, animals do not freeze to the CS- nor the CS+ but freeze to the shock context. We have included this new experiment in the manuscript and revised the text to clarify this point.

7) Throughout the manuscript, the rats' fear responses to the CS- are quite variable. Indeed, in each case, there appears to be a bimodal distribution (see Figs. 1A, 1B, 2C, 3A, 5A). The highly variable nature of the freezing in response to the CS- weakens the argument that the neutral and fearful events are "linked" in all animals. It would be interesting to know what factors, at the neural level, for instance, can explain such variability.

We completely agree. In several experiments, we observed a bimodal distribution, which led us to investigate whether the response to the CS- was correlated with the response to the CS+. Indeed, we found a significant correlation, which strongly supports our hypothesis that these two events are linked. Furthermore, we agree on the importance of exploring the factors contributing to this variability. Our paradigm is particularly well-suited for such investigations. However, this falls beyond the scope of the present study. That said, we encourage you to follow our upcoming research, as we plan to examine the hypothesis that variability in the endocannabinoid system may play a role in this phenomenon.

8) On p. 6, the authors state "One group of animals freezes to both stimuli ... whereas a second group does not freeze to either stimulus." The second group does freeze, just less. This needs reworded.

Thank you for pointing this out. We have rephrased the text accordingly.

9) On p. 6, the authors suggest that animals in the 30-minute interval "generalize" their responses, choosing to either generalize freezing or not freezing. In the previous paragraph, the authors argue that the association effect is not due to an overall generalization of fear. The jargon in paragraphs one and two can get mixed. This needs more careful attention to wording. Moreover, the relationship

between CS+ and CS- freezing is correlational; no causal inferences can be made here.

We apologize for using generalization here given that, yes, we agree it is not generalization but rather association. We have clarified the text accordingly.

10) It is unclear what novel information is being presented by the results displayed in Fig. 3. The inclusion of a third event (with a light stimulus) prevents the generalization made between the neutral and fearful events. However, this seems like a simple case of memory interference. Plus, the added stimulus is from a different sensory modality.

Thank you for raising this point. One could say that yes, there is an interference of light in the acquisition of fear to the CS-. However, there is no effect of light in CS+ acquisition, so no interference, and there is a positive effect of light in the CS+ response. By dissociating the CS+ from the CS- the light event blocked the competitive effect of the CS-. In this case, we think competition is more appropriate given that one of the stimuli “wins”. In the case where the light event is presented after the CS+/CS- association (Supplementary Figure 5B) then interference may be more correct. In this case, the light resulted in the reduction of the response of both cues, thus interfering with both and no winners exist. We have clarified this in the discussion.

11) It would be useful to know if the response to the CS- is different from the response to the CS+ throughout the experiments. In some cases, fear to the CS- appears to be significantly less than fear to the CS+, even though it is greater than "pre".

In all figures, the statistically significant differences are depicted except for the comparison between the CS- in T1 and T2 in Figure 5D (please see comment below). If there is no graphical representation in the figures then no statistically significant differences exist. Additionally, please find all P values in the statistical table available in the manuscript.

12) In Fig. 5D, it would be useful to know if CS- under T1 is different from CS- under T2.

Thank you for this comment. We have not directly compared differences between the CS- values across time. This is now added to the statistical table and in the figures.

Reviewer #2

1. The authors discuss memory generalization, but an alternative explanation could be that some mice generally fail to form memories when two events are spaced 30 minutes apart. To address this, I suggest conducting an additional test 24 hours or more after the 30-minute interval test to verify whether the memory persists in animals that exhibit reduced freezing. This phenomenon should also be discussed in greater detail, considering the possibility that presenting two events with a 30-minute interval might impair memory formation itself (resulting in reduced freezing in both contexts). The discussion briefly mentions this possibility (“in some animals, the neutral event can interfere with the fear memory acquisition of the aversive event”), but the mechanisms and alignment with the proposed hypothesis need further elaboration. Understanding this baseline phenomenon without manipulations is crucial for interpreting the circuits and molecular mechanisms involved and trying to reconciling the differing timings reported in previous studies.

Thank you for this comment. We hypothesize that if events occur within the 30-minute interval the events are linked. However, this linkage can lead in some animals to a high cue-evoked response, to both auditory cues, and in some animals to a low cue-evoked response again to both cues. This association occurs during training and it is observed 24 hours later in the test session (please see paradigm diagram add in figures). We performed two additional experiments to further characterize the time course of the 30-minute association. In this first experiment, we assessed whether at 3 hours after training the association is already present (Supplementary Figure 2A). We observed that if tested at 3 hours, animals display freezing to both CS- and CS+, showing a correlation between responses to the auditory cues. These animals were further tested at 24 hours and as seen in the reactivation experiment, the response to the CS- decreased and it was no longer different from baseline responses. This experiment shows that CS+ and CS- association occurs during training. In the second experiment, we tested whether this association was enduring (Supplementary Figure 2C). We observed that animals tested at 72 hours freeze to both CS- and CS+, showing that this association is not transient. Again, a positive correlation was observed between CS- and CS+ evoked freezing responses.

2. The sentence “activating thalamic inputs reduces the time window of memory cooperation” seems to contradict the study’s findings. Please clarify this point.

Thank you for raising this point. You are absolutely right; the sentence does not make sense. We have modified it.

3. When CB1 was inhibited during a 1-hour delay, the authors observed increased memory association. However, no mice exhibited reduced freezing in these conditions as seen in the 30-minute control experiments. Could this reflect heightened anxiety? Although the discussion addresses this possibility briefly, I recommend performing anxiety tests to isolate the drug’s specific effects.

Thank you for raising this point. We agree that inhibiting CB1R in the amygdala results in increased arousal and overall responsiveness of the amygdala. Although the drug is infused only in the time interval between the CS- and CS+ cue presentation, given the expression of CB1R in excitatory and inhibitory neurons, this experiment, although informative, is rather unspecific. Indeed, there is an increase in baseline freezing (Pre) interval which is consistent with an increase in arousal.

4. The authors state that “thalamic activation increases the time window... but does not determine the direction of such interaction,” but later claim that their results support “the hypothesis that thalamic activation determines whether animals associate or discriminate events.” These statements seem contradictory and require rephrasing for consistency.

Thank you for raising this point. These statements are not contradictory. The hypothesis is that the activity of the thalamic input determines the time window during which events can be associated. If we increase thalamic activity (e.g. by chemogenetic manipulation of MGm projections) we increase the time window favoring association. However, association does not mean more cue-evoked responses e.g. more fear. The association can lead to reduced fear of both cues and thus increasing association, by increasing thalamic activation, does not determine the direction of response. The text in the discussion has been revised to clarify this point.

5. The authors claim consistency between their findings and the 2016 Rashid study, but Rashid used a minimum interval of 90 minutes. This discrepancy makes the comparison unclear, and I recommend rephrasing this discussion for clarity.

Thank you, the text has been revised to clarify this point.

Minor Comments

1. *Page 6: Figure 1B is not cited in the text.*

Thank you, we have rephrased the text accordingly.

2. *Figure Titles: Adding titles to individual panels in all figures would improve readability (e.g., for Figure 1, titles such as "7.5 min interval," "30 min interval," and "1h interval" for panels A, B, and C, respectively).*

We have reorganized the figures to increase clarity.

3. *Figure 2A Annotations: References or labels for the targeted brain regions visible in the figure would be helpful.*

Thank you for the suggestion, this has been added.

4. *Color Coding: The same three colors are used to represent both contexts and intervals, which can cause confusion. I suggest adopting a different color scheme.*

Thank you for the suggestion, we have reorganized the figures to increase clarity.

5. *Figure 3B Clarity:*

The distinction between "pre" and "pre group" is unclear. Specify when the light group is analyzed (e.g., during light exposure or during the CS+ phase).

In the light experiment, animals are tested twice, 24 hours after training are tested for auditory cue-evoked responses, and 48 hours after training, return to be tested for light-evoked responses (please see the diagram added to the figures).

Reviewer #3

Given that transfection can be transsynaptic, the authors need to rule out this possibility to support the main conclusions on thalamic input to the lateral amygdala.

Appropriate control groups are needed. In Fig. 2, Gq with vehicle control is missing. Gq with CNO in the lateral amygdala is also missing, which would be critical to support the sufficiency for this pathway in CS- freezing.

Thank you for raising these points which we believe are linked. Regarding the trans-synaptic

expression of our constructs, we do not see cell bodies transfected except for the site of viral injection. Since most of our experiments rely on the targeting of the MGm thalamic nuclei, this leads to the transfection of axonal projections from MGm neurons to the amygdala and other brain areas (please see images below) but no cell bodies are labeled. Regarding control groups, we have now added a control group of Gq saline in Figure 2. We would like to clarify that we do not propose that the thalamic input is sufficient to form an associative memory that subsequently results in a response to the neutral CS- event. We propose that modulating the thalamic input alters the time window during which memory association can occur. Indeed, you are correct that there is no experiment designed to exclude the role of the cortical input in the fear memory acquisition but rather that increasing the activation of these inputs does not alter the time window of association. Thalamic axons from the MGm nuclei project to other areas than the amygdala, including the auditory cortex and the entorhinal cortex which are adjacent to the amygdala and would be activated even with local infusion of CNO (please see images below).

In Fig. 3, the hypothesis for the light stimuli in the competitive study can be arbitrary and more data reporting and control studies will be needed to make the competition concept convincing. How to rule out the possibility that CS- was associated with the recent event with light which was safe? It requires a control group with context 2 and no light in between CS+ and CS- to show that this effect is specific to light and not due to a novel and safe context.

Thank you for raising this point and we apologize if it was not clear. The control group depicted in Figure 3A was done exactly in that manner, that is, animals go into context 2 but no light is presented there. In the revised figures, we added a graphical representation of the protocol used to link the protocol with the corresponding data for clarification.

The freezing to CS- and to the context 30min after conditioning should be provided for the group in Fig. 3A and freezing to light, CS-, and contexts soon after conditioning should be provided for the group in Fig. 3B. These will enable understanding if association or generalization had already happened soon after conditioning.

Thank you for raising this point. Regarding the timeline of association, as stated before, we performed an additional experiment where animals were tested 3 hours after training and we observed that the association was already present (Supplementary Figure 2A). As shown in this experiment and in the reactivation experiment (Figure 5) animals cannot be re-tested given that the responses to the CS- significantly reduce if animals are tested with either cue. Regarding responses to context, this was not tested except for the new experiment depicted in supplementary figure 1B. In this new setting, the CS+ was omitted during the aversive event and thus, animals show no response to both auditory cues. They do show an aversive response to the trained context. In all our experiments animals were always tested for cued responses in novel contexts to minimize contamination of context in cue-evoked responses.

As Fig. 3D showed that Light did not lead to significant freezing on the next day, why is it not associative like the CS- and how did it lead to 'competition' when itself was not sufficient as a potential associative stimulus?

Thank you for raising this point. As stated above, electrophysiological characterization of MGm neurons (thalamic) shows that these cells are multimodal whereas MGv cells are not. Thus, using a

stimulus from a different modality, in this case, a light, we aimed to more selectively stimulate the MGm.

In Fig. 4, the extent of transfection in LA should be made clear in relation to fiber optic cannula tips in A'/A''.

We would like to clarify that the LA is not directly transfected, we apologize if this was not clear. In all experiments where we probe thalamic inputs, including optogenetic experiments, in the amygdala we find axonal projections labeled by transfection of MGm/PIN thalamic nuclei. The pattern is very similar across animals (please see images below) and spread throughout all amygdala areas. Fiber optic cannula tips are localized slightly above the amygdala as depicted in Figure 4A.

The specific time point of optogenetic stimulation between CS+ and CS- should be clearly described. Since optogenetic stimulation is very transient and activities would likely return to normal during CS-, how did this lead to observed disruption?

Thank you for raising this point. Optogenetic stimulation occurred for 15 minutes during the 30-minute interval between the CS+ and the CS- (30 pulses at 0.033Hz). Optogenetic stimulation started 7.5 minutes after the transition to context 2 and was terminated 7.5 minutes before CS- presentation. We agree that when the CS- is presented, the activity of thalamic fiber has returned to baseline. Similarly, in the light competition experiment (Figure 3) the activity induced by the light presentation has also returned to baseline when the CS- is presented (same interval as for the optogenetic experiment). We have updated Figure 4 and updated the methods section to clarify this.

Brain regions, contours, and scale bars need to be superimposed in all histology figures to enable visualization of the transfection areas. A schematic image is needed to summarize variation in transfected areas across rats.

Thank you for the suggestion, this has been added now.

More information is needed to enable unbiased conclusions and reproduction. For example, are all contexts novel and counterbalanced? How are contexts A-D matched to contexts 1-2, 1-3, or 1-4? Photos of contexts A-D should be provided.

We apologize for this mistake. Indeed, we use four different and novel contexts that are counterbalanced across experiments. We have revised the methods to clarify this point. Also, please find below photos of the contexts used.

It seems that the auditory stimuli used as CS- and CS+ were not counterbalanced. Would this affect the generality of the observed phenomenon?

Thank you for raising this point, yes, the auditory cues were not used in a counterbalanced manner. This is because animals do not respond equally to high (or higher) and frequency low-frequency sounds. We decided that it would be preferable to use a frequency low-frequency tone as CS+ and maintained this selection throughout this work. We have now added an experiment in which we

switch the auditory cues used as CS- and CS+ (please see diagram of paradigm in supplementary figure 1). We observed that for the 7.5 interval, as seen before, using opposite frequency sounds as auditory cues results in associative learning, with animals responding to both cues.

To show that the CS- freezing reflects associative learning, more evidence beyond the CS- omission control in sup. Fig. 1 will be needed.

To rule out that animals are associating the neutral auditory cue (CS-) to the shock, we tested whether omitting the CS+ during training led to a fear response 24 hours later. In this case, animals are exposed to the first context where the CS- is presented and 7.5 minutes later move to a different context and receive 5 foot-shock without the CS+ (Supplementary figure 1B). In this situation, in the test session, animals do not freeze to the CS- nor the CS+ but freeze to the shock context. This shows that an association between the two auditory cues occurred during training. We have included this new experiment in the manuscript and revised the text to clarify this point.

The procedure diagrams on top of each figure should make it clear as to how many groups were done and described in data figures below (as in Fig. 5).

Thank you for the suggestion, this is now added.

Gq and Chr2 transfection areas

Gq + CNO 1h interval n=10

Open Vector + CNO 1h interval n=8

Gq + CNO 1h interval no CS- n=8

Gq + saline 1h interval n=10

Cortical Gq n=9

Cortical Open vector n=7

Amygdala_Chr2 Gq n=9

Amygdala_Chr2_open vector n=7

Amygdala

Photos of contexts